# Cryo-EM structures of the autoinhibited *E. coli* ATP synthase in three rotational states

**Meghna Sobti[1], Callum Smits[1], Andrew SW Wong[2], Robert Ishmukhametov[3], Daniela Stock[1,4], Sara Sandin[2,5], Alastair G Stewart[1,4]***

[1]Molecular, Structural and Computational Biology Division, The Victor Chang Cardiac Research Institute, Darlinghurst, Australia; [2]NTU Institute of Structural Biology, Nanyang Technological University, Singapore, Singapore; [3]Department of Physics, Clarendon Laboratory, University of Oxford, Oxford, United Kingdom; [4]Faculty of Medicine, The University of New South Wales, Sydney, Australia; [5]School of Biological Sciences, Nanyang Technological University, Singapore, Singapore

**Abstract** A molecular model that provides a framework for interpreting the wealth of functional information obtained on the *E. coli* F-ATP synthase has been generated using cryo-electron microscopy. Three different states that relate to rotation of the enzyme were observed, with the central stalk's ε subunit in an extended autoinhibitory conformation in all three states. The $F_o$ motor comprises of seven transmembrane helices and a decameric c-ring and invaginations on either side of the membrane indicate the entry and exit channels for protons. The proton translocating subunit contains near parallel helices inclined by ~30° to the membrane, a feature now synonymous with rotary ATPases. For the first time in this rotary ATPase subtype, the peripheral stalk is resolved over its entire length of the complex, revealing the $F_1$ attachment points and a coiled-coil that bifurcates toward the membrane with its helices separating to embrace subunit *a* from two sides.

*For correspondence: a.stewart@victorchang.edu.au

**Competing interests:** The authors declare that no competing interests exist.

## Introduction

In most cells, the bulk of ATP, the principal source of cellular energy, is synthesized by ATP synthase. This molecular generator couples ion flow across membranes with the addition of inorganic phosphate (Pi) to ADP thereby generating ATP (*Iino and Noji, 2013*; *Stewart et al., 2014*). Most bacteria, including *Escherichia coli* have only one type of rotary ATPase, referred to as F-type ATPase. Like the analogous complexes in other kingdoms, it is based on two reversible motors, termed $F_1$ and $F_o$ (*Negrin et al., 1980*), connected by central and peripheral stalks (*Wilkens and Capaldi, 1998a*) (*Figure 1*). The $F_o$ motor spans the membrane converting the potential energy of the proton motive force (pmf) into rotation of the central stalk that in turn drives conformational changes in the $F_1$ catalytic sites.

The $F_o$ motor is constructed from subunits *a*, *b* and *c* (*Figure 1*). Subunit *c* assembles into a ring, thought, in *E. coli*, to have decameric stoichiometry (*Jiang et al., 2001*; *Ballhausen et al., 2009*; *Düser et al., 2009*; *Ishmukhametov et al., 2010*), whereas subunits *a* and *b* associate to form a helical bundle adjacent to this ring. Recent sub nanometer electron cryo-microscopy (cryo-EM) reconstructions of F-type (*Allegretti et al., 2015*; *Zhou et al., 2015*; *Kühlbrandt and Davies, 2016*; *Hahn et al., 2016*) and the analogous V- and A-type ATPases (*Zhao et al., 2015*; *Schep et al., 2016*) as well as a low-resolution crystal structure of *Paracoccus denitrificans* F-ATPase (*Morales-Rios et al., 2015*) are consistent with a two half-channel mechanism for the generation of rotation within the membrane (*Vik and Antonio, 1994*; *Junge et al., 1997*). All structures confirm that a

**eLife digest** ATP synthase is a biological motor that produces a molecule called adenosine tri-phosphate (ATP for short), which acts as the major store of chemical energy in cells. A single molecule of ATP contains three phosphate groups: the cell can remove one of these phosphates to make a molecule called adenosine di-phosphate (ADP) and release energy to drive a variety of biological processes.

ATP synthase sits in the membranes that separate cell compartments or form barriers around cells. When cells break down food they transport hydrogen ions across these membranes so that each side of the membrane has a different level (or "concentration") of hydrogen ions. Movement of hydrogen ions from an area with a high concentration to a low concentration causes ATP synthase to rotate like a turbine. This rotation of the enzyme results in ATP synthase adding a phosphate group to ADP to make a new molecule of ATP. In certain conditions cells need to switch off the ATP synthase and this is done by changing the shape of the central shaft in a process called autoinhibition, which blocks the rotation.

The ATP synthase from a bacterium known as *E. coli* – which is commonly found in the human gut –has been used as a model to study how this biological motor works. However, since the precise details of the three-dimensional structure of ATP synthase have remained unclear it has been difficult to interpret the results of these studies.

Sobti et al. used a technique called Cryo-electron microscopy to investigate the structure of ATP synthase from *E. coli*. This made it possible to develop a three-dimensional model of the ATP synthase in its autoinhibited form. The structural data could also be split into three distinct shapes that relate to dwell points in the rotation of the motor where the rotation has been inhibited. These models further our understanding of ATP synthases and provide a template to understand the findings of previous studies.

Further work will be needed to understand this essential biological process at the atomic level in both its inhibited and uninhibited form. This will reveal the inner workings of a marvel of the natural world and may also lead to the discovery of new antibiotics against related bacteria that cause diseases in humans.

four-helix bundle of subunit *a*, inclined by 20–30° to the membrane plane, forms a crucial structural component. In this mechanism, protons from the bacterial periplasm access a conserved negatively charged carboxylate in subunit *c* (Asp61 in *E. coli* [*Hoppe et al., 1982*]) through an aqueous half channel at the subunit *a/c* interface (*Steed and Fillingame, 2008*). Neutralizing this carboxylate enables the *c*-ring to rotate within the hydrophobic membrane and to access a second aqueous half channel that opens to the cytoplasm into which the protons are released (*Pogoryelov et al., 2010*). A conserved arginine residue in helix-4 of subunit *a* (Arg210 in *E. coli*) prevents the *c*-ring rotating in the opposite direction and short-circuiting of the system (*Lightowlers et al., 1987*; *Cain and Simoni, 1989*; *Mitome et al., 2010*). The sequential binding of protons in combination with thermal fluctuations generates rotation within the complex in a manner akin to a turbine (*Oster and Wang, 1999*; *Pogoryelov et al., 2010*; *Aksimentiev et al., 2004*). The torque generated in the $F_o$ motor is then transferred to the $F_1$ motor by the central shaft consisting of subunits γ and ε (*Wilkens et al., 1995*). The N- and C-termini of subunit γ form a curved coiled-coil that extends into the central cavity of $F_1$.

The $F_1$ motor is the chemical generator in which ATP is synthesized. The motor comprises a ring of three heterodimers, each containing an active site at the interface of subunits α and *β*. Within the $F_1$ motor, each α*β* dimer has a different conformation at any point in time and can be either empty, bound to ADP and Pi, or bound to ATP (open, half-closed, closed) (*Abrahams et al., 1994*; *Yoshida et al., 2001*). These different catalytic states relate to the position of the curved coiled-coil of subunit γ in the central stalk, which drives the conformational changes associated with catalysis. To enable the central stalk to rotate relative to the $F_1$ α*β* heterodimers, the $F_o$ and $F_1$ motors need to be coupled. This coupling is mediated by the peripheral stalk that is constructed from subunits *b* and δ (*Figure 1*). Subunit *b* forms an amphipathic homodimeric coiled-coil that spans the periphery

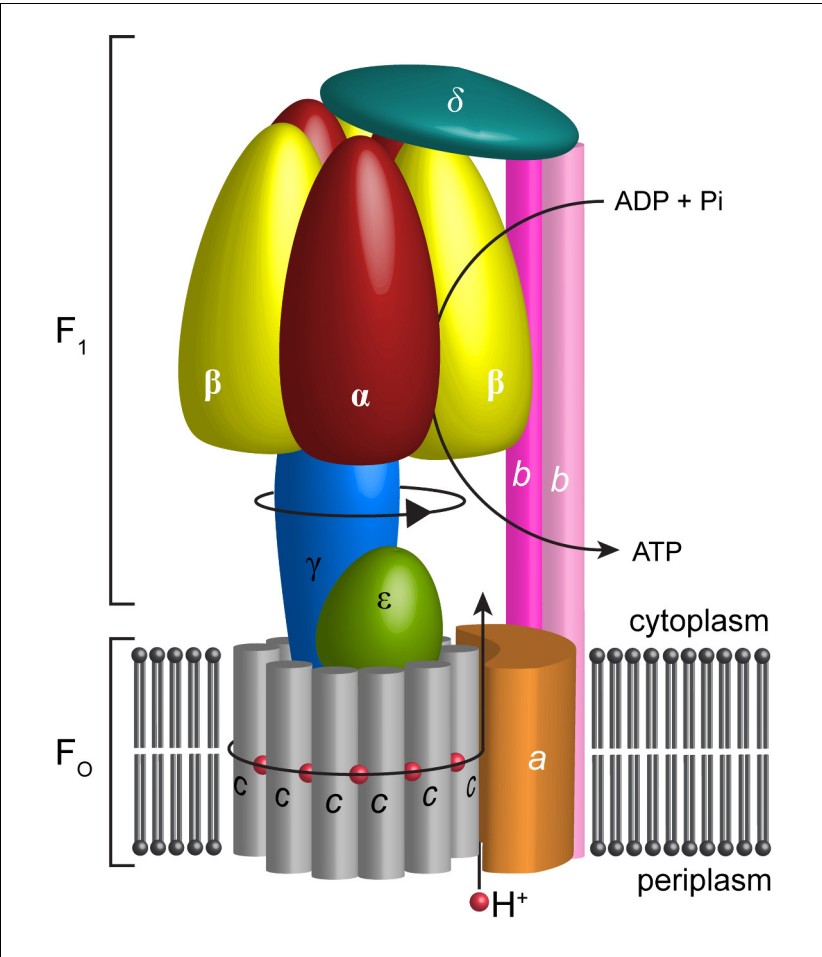

**Figure 1.** Schematic illustration showing the arrangement of subunits in *E. coli* F-ATPase. Subunits α in red, *β* in yellow, γ in blue, ε in green, *c* in grey, *a* in orange, *b* in magenta or pink, and *δ* in teal. The proton path and ATP synthesis are labeled accordingly.

of the complex linking subunit *a* with subunit α, whereas subunit δ provides additional coupling of the C-termini of the *b* and α subunits (*Carbajo et al., 2005*; *Wilkens et al., 2005*).

The bacterial F-ATPase can also function in reverse, employing ATP hydrolysis to generate a proton gradient across the membrane when needed (*von Ballmoos et al., 2008*). In vivo, subunit ε is believed to change conformation in an ATP-dependent manner to prevent rotation of the complex (*Capaldi et al., 1992*; *Rodgers and Wilce, 2000*; *Yagi et al., 2007*; *Imamura et al., 2009*) thereby conserving ATP when its concentration is low. This regulatory function is mediated by the C-terminal domain of subunit ε (εCTD) that, when not bound to ATP, opens to an extended conformation and inserts into the αβ heterodimers. The crystal structures of both the *E. coli* and *Bacillus* PS3 $F_1$ motors in this autoinhibited state show the εCTD intercalating into the αβ heterodimers (*Cingolani and Duncan, 2011*; *Shirakihara et al., 2015*). However, in each structure, the $F_1$ motor had been captured in a different conformation (*E. coli* – half-closed, closed, open [*Cingolani and Duncan, 2011*] and *Bacillus* PS3 – open, closed, open [*Shirakihara et al., 2015*]) which could either relate to inter species differences or crystal contacts and crystallization conditions.

The crystal structure of the F-ATPase from *P. denitrificans* (*Morales-Rios et al., 2015*), that is closely related to *E. coli* (38% sequence identity over all subunits), shows a similar overall architecture to bovine $F_1F_o$ ATP synthase as well as to the main features of A/V ATPases. However, it is inhibited by the ζ-protein rather than by subunit ε, which is generally employed by bacterial

F-ATPases for this purpose. Moreover, this crystal structure shows only one conformation of the rotary catalytic cycle.

Here, we present three cryo-EM maps along with molecular models of *E. coli* F-ATPase in its auto-inhibited state, determined to resolutions of 6.9, 7.8, and 8.5 Å. In all three reconstructions, the εCTD is in an extended conformation, stabilizing an overall $F_1$ motor conformation similar to that seen in the thermophilic *Bacillus* PS3 $F_1$ ATPase structure. Density for the peripheral stalk extends the entire length of the complex and its coiled-coil bifurcates towards the N-terminus to enter the membrane as two separate helices that clamp the *a* subunit to the *c* ring. Moreover, our maps allowed us to interpret the complete $F_1$-delta interface, showing the three α subunit N-termini in distinct orientations. Each map also confirmed the *c*-ring stoichiometry to be decameric, which to date has been only characterized by crosslinking and single molecule analyses. We used our maps in combination with published crosslinking and mutagenesis information to generate a molecular model of the complex in three states. These models provide crucial structural information on a key complex that extends our understanding of the mechanism of rotary ATPases in general, together with information on the bacterial ATP synthase, which is seen as an important antimicrobial target in organisms related to *E. coli* such as *Mycobacterium tuberculosis* (*Ahmad et al., 2013*).

## Results

### Complete molecular models of three different F-ATPase conformations

Cysteine-free *E. coli* F-ATPase, as described in *Ishmukhametov et al. (2005)* where all 10 cysteines were replaced with alanines and a His-tag introduced on the *β* subunit, was solubilized in digitonin detergent and purified as described in the Materials and methods. This procedure provided pure protein (*Figure 2—figure supplement 1*) capable of ATP hydrolysis-driven proton pumping upon reconstitution into proteoliposomes (*Figure 2—figure supplement 1*). N,N'-dicyclohexylcarbodii-mide (DCCD), a compound which selectively modifies Asp61 of subunit *c* at 50 μM (*Pogoryelov et al., 2010*) completely abolished proton pumping (*Figure 2—figure supplement 1B*) and inhibited 90% of ATPase activity of isolated protein (*Figure 2—figure supplement 1C*). Such inhibition indicates coupling between the $F_1$ and $F_o$ motors (*Cook et al., 2003*; *Peskova and Nakamoto, 2000*; *Tsunoda et al., 2000*).

Protein was further examined by cryo-EM without addition of nucleotides. 395,140 particles were picked, of which 216,711 were used in refinement. Three different conformations of the complex were identified using 3D classification in RELION (*Scheres, 2012*). The particles in each subset were then refined to generate sub-nanometre reconstructions, to a resolution of 6.9, 7.8 and 8.5 Å (*Figure 2A–C*, and *Figure 2—figure supplements 2* and *3*). In these three conformations, the central stalk was progressively rotated 120° relative to the peripheral stalk.

Even though the resolution of the reconstructions varied throughout the complex, it was sufficient to resolve individual helices. Additional density of the N-terminal His-tag of the *β* subunit, as well as helical and *β* sheet patterns observed in parts of the map in the $F_1$ motor region illustrate the high quality of the maps, with the *c*-ring density being poorest (*Figure 2—figure supplement 4*). Local resolution estimates showed the region corresponding to the $F_1$ motor to be of highest quality, the $F_o$ motor with moderate detail and, the detergent micelle being clearly the worst region of the map (*Figure 2—figure supplement 5*). Docking of high-resolution crystal and NMR models of different components into the maps followed by manual building and refinement enabled virtually complete molecular models of the three different states to be built (*Figure 2D–F*), with varying quality of the docked structures as indicated in *Figure 2—figure supplement 6*. The positions of the Cys-Ala mutants are depicted in *Figure 2—figure supplement 7*.

The cryo-EM maps provided novel insights into the architecture and function of the *E. coli* F-ATPase. Thus, although its overall architecture was similar to that of F-ATPase from *P. denitrificans* (*Morales-Rios et al., 2015*) and $F_1F_o$ ATP synthases from *Bos Taurus* (*Zhou et al., 2015*), *Yarrowia lipolytica* (*Hahn et al., 2016*) and *Polytomella* (*Allegretti et al., 2015*), with the catalytic $F_1$ motor attached to a proton powered membrane $F_o$ motor and single central and peripheral stalks, differences in the individual motors and peripheral stalk were apparent. Comparison with the membrane-embedded motors from other sub nanometre cryo-EM maps indicated the *E. coli* F-ATPase had a simpler stator architecture, containing only seven helices in the *a* and *b* subunits rather than the

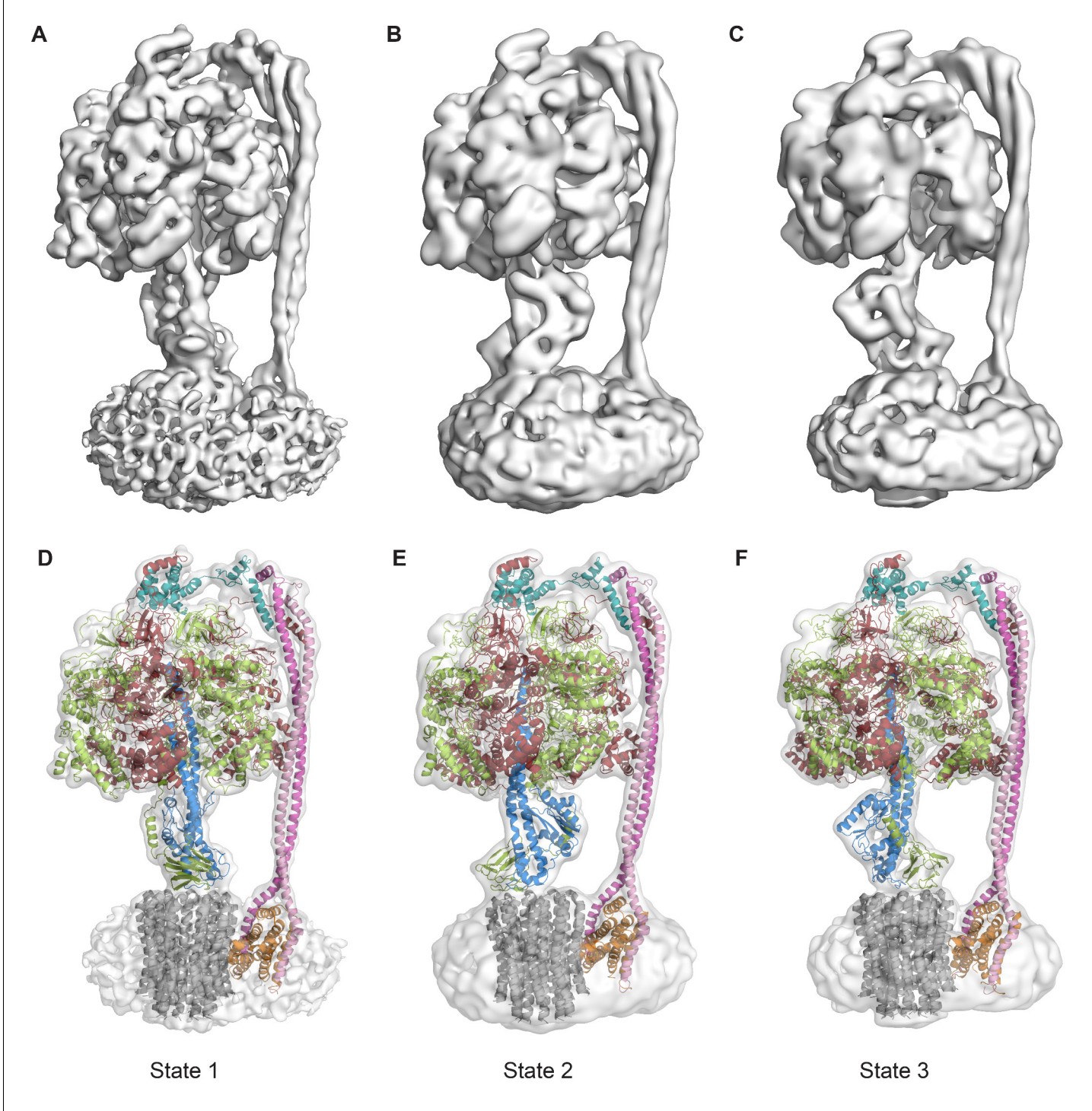

**Figure 2.** The three states of the autoinhibited *E. coli* F-ATPase. (A–B) Cryo-EM maps shown as surface representation, states 1, 2 and 3, respectively, resulting from rotation of the central stalk by 120°. (D–F) Molecular models built into the cryo-EM maps shown as cartoon representation. Subunits α in red, β in yellow, γ in blue, ε in green, c in grey, a in orange, b in magenta or pink and δ in teal.

The following source data and figure supplements are available for figure 2:

**Source data 1.** Data collection and image processing statistics.

**Figure supplement 1.** Characterization of *E.coli* $F_1F_o$ ATP synthase used for cryo-EM.

*Figure 2 continued on next page*

*Figure 2 continued*

**Figure supplement 2.** cryoEM analysis.

**Figure supplement 3.** Flowchart describing cryoEM data analysis.

**Figure supplement 4.** Examples of the electron density map of State 1, to highlight strengths and weaknesses.

**Figure supplement 5.** Local resolution map of State 1.

**Figure supplement 6.** Quality of the models built into the state one cryoEM map.

**Figure supplement 7.** Position of the natural cysteines in *E. coli* $F_1F_o$.

**Figure supplement 8.** Transmembrane architecture of (**A**) *E.coli*, (**B**) *P. denitrificans*, (**C**) *Y. lipolytica*.

**Figure supplement 9.** Comparison of peripheral stalk position between the three states; diagrams on left depict part of complex that each state is superposed to.

**Figure supplement 10.** FSC curves showing the effects of masking on the refined map, with the gold-standard, corrected FSC curve (black), FSC of the unmasked map (green), FSC of the masked map (blue), and FSC of the phase-randomized masked map (red).

eight seen in mitochondrial F-type ATP synthase (*Figure 2—figure supplement 8*), consistent with labeling approaches (*Wada et al., 1999*). This difference suggested that the extra helices present in other rotary ATPase subtypes could have additional functions such as the dimerization seen in mitochondria (*Hahn et al., 2016*).

A movie generated by interpolation between the three states (*Video 1*) indicated that the $F_1$ motor rocks or wobbles during the catalytic cycle (*Kinosita et al., 2000*; *Stewart et al., 2012*) as previously predicted, although of course the structures described here do not represent the complex in its uninhibited active synthesizing form. Two pivot points, one near the peripheral stalk/$F_o$ interface (~$b$Arg36) (*Welch et al., 2008*) and one near the peripheral stalk/$F_1$ interface (~$b$Gln106), enabled the stalk to accommodate this eccentric movement of $F_1$ (*Figure 2—figure supplement 9*).

## Subunit δ interaction with subunit b dimer and all three α subunits

The maps showed a long right-handed coiled-coil dimer generated by the two *b* subunits of the peripheral stalk together with the globular δ subunit that anchors them to the catalytic head (*Figure 3A*). The quality of the map was sufficient to enable almost the entire of the δ subunit to be built as a polyalanine model (*Figure 2D–F*), whereas previous structural information was limited to the N-terminal domain (*Wilkens et al., 2005*). Interestingly, the peripheral stalk contacted all three α subunits via their N-terminal helices, but did so asymmetrically employing three different interfaces with each α subunit (*Figure 4*). Although the resolution of the map was insufficient to assign the precise interface, the binding of the peripheral stalk to three anchor points in different geometries would provide a molecular key that would result in the δ subunit binding in a single orientation across the top of the symmetrical αβ heterodimers.

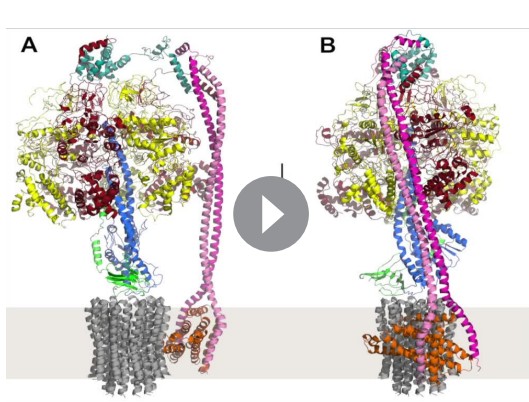

**Video 1.** Interpolation between States 1, 3 and 2 to simulate ATP synthesis by *E. coli* F-ATPase. **A** and **B** are rotated 90° about the y-axis.

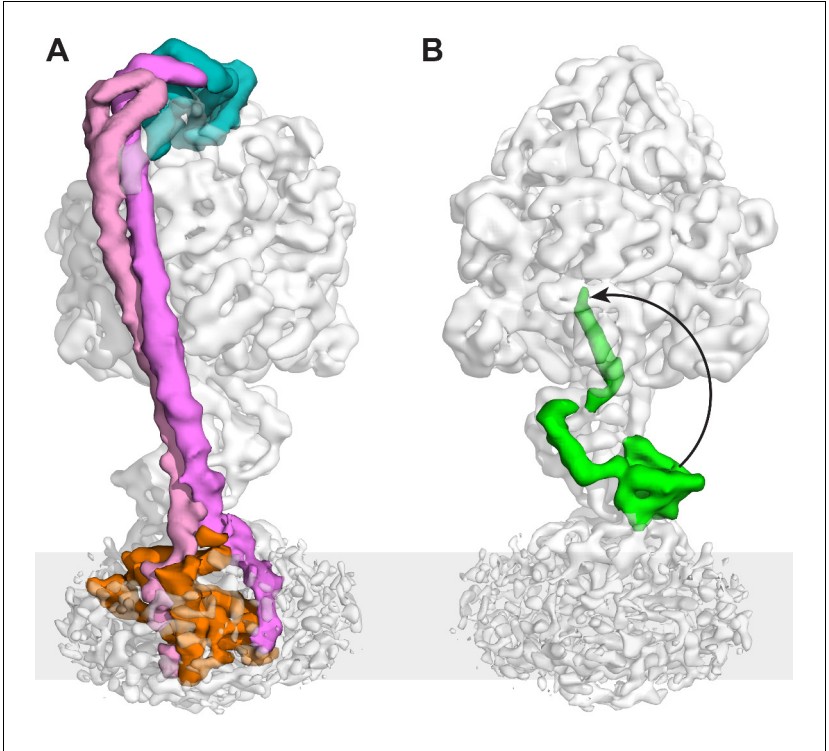

**Figure 3.** The peripheral and central stalks of *E. coli* F-ATPase. (**A**) The peripheral stalk is comprised of a globular head (subunit δ in teal) and a homodimeric coiled-coil (subunits *b* in pink and magenta) that bifurcates at the membrane interface to brace subunit *a* (orange). (**B**) The εCTD is in an extended conformation, inhibiting the enzyme from rotating. The arrow depicts the extended vs closed conformation of subunit ε.

The following figure supplements are available for figure 3:

**Figure supplement 1.** Fitting of the of the autoinhibited *E.coli* F$_1$-ATPase crystal structure (pdb 3oaa) into the State one cryoEM map of *E. coli* F-ATPase.

**Figure supplement 2.** Stimulation of ATP hydrolase activity of isolated F$_1$F$_o$ by 0.4% LDAO.

## The peripheral stalk bifurcates into the membrane

The *b* subunits formed a homodimeric coiled-coil that spaned almost the entire complex (212 of 232 Å) with their N-termini bifurcating just above the membrane to generate two separate helices within the membrane (*Figure 3A*). This was unexpected, albeit reminiscent of the yeast F-type ATP synthase dimer (*Figure 2—figure supplement 8C*), where subunit eight is an evolutionary derivative of the bacterial *b* subunit (*Hahn et al., 2016*). Furthermore, NMR analysis of the transmembrane domain of *E. coli* F-ATPase *b* subunit (*Dmitriev et al., 1999*) showed a helical structure that was interrupted by a rigid 20° bend at residues 23–26 that result in a structure consistent with the *b* subunit bifurcation.

## Inhibition of the *E. coli* F-ATPase by central stalk subunit ε

All three reconstructions showed the complex in its autoinhibited state, with clear density for the εCTD extending deep into the central cavity of the F$_1$ enzyme (*Figure 3B*). Fitting of the *E. coli* α$_3$β$_3$γε crystal structure (*Cingolani and Duncan, 2011*) into our cryo-EM maps showed that the β1 subunit had adopted a different more open conformation (*Figure 3—figure supplement 1*). In the above crystal structure, the εCTD contacts more subunits in F$_1$ (α1, α2, β1, β2 and γ) compared to our cryo-EM reconstructions, where it contacted fewer subunits (α1, α2, β2 and γ). The conformation of our cryo-EM structure was more similar to that seen in the *Bacillus* PS3 structure

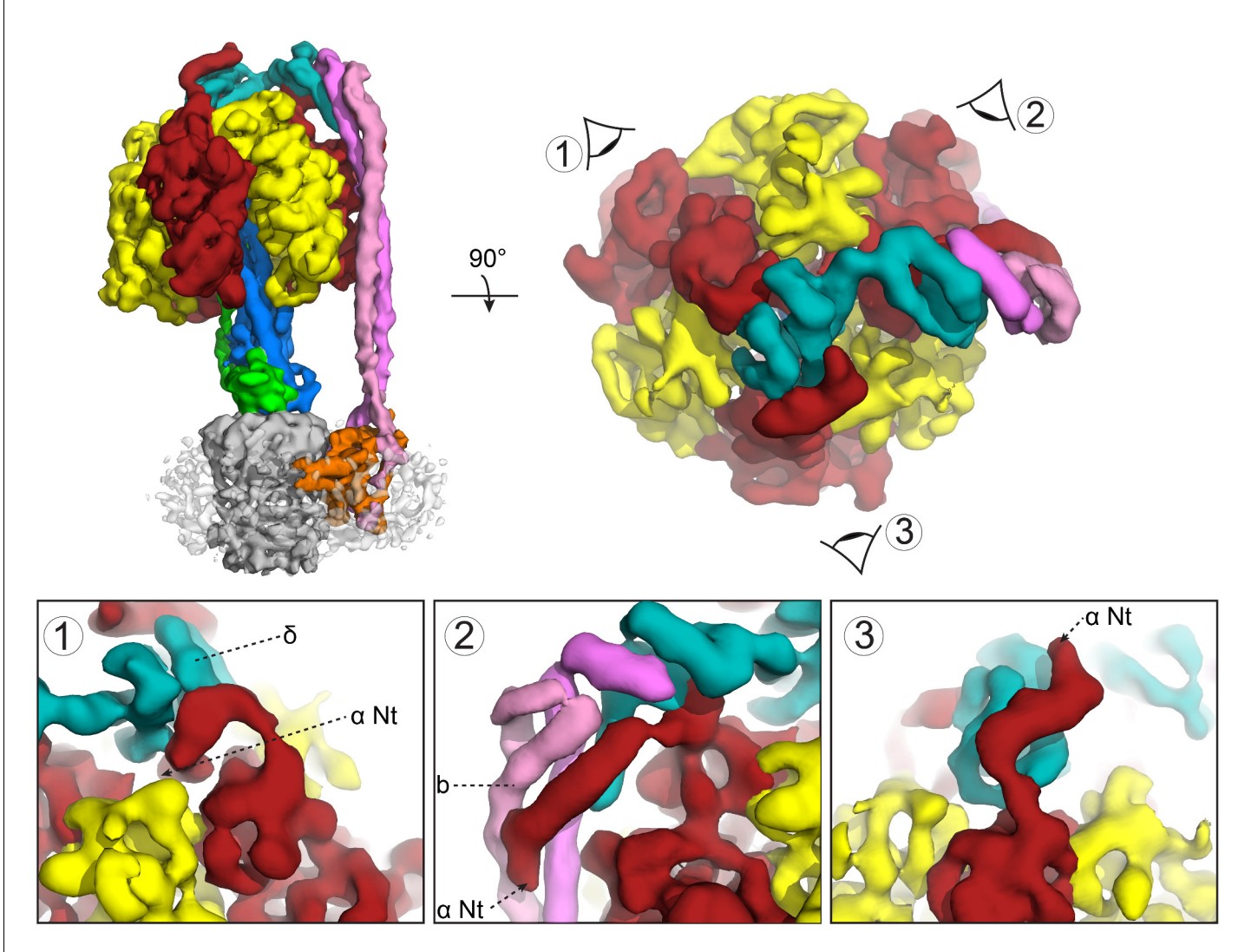

**Figure 4.** Subunit δ and peripheral stalk attachments to the α subunits. Top panel; left, the segmented cryoEM map viewed from the side and right, viewed from above with the orientation of views 1, 2 and 3 depicted. Bottom panel; detailed views of the three attachment points labeled 1, 2 and 3, with δ in teal, b in pink and magenta and α in red.

(*Shirakihara et al., 2015*), where it is proposed to be in an 'open, closed, open' conformation (*Figure 5*).

To ascertain the enzymatic state of the $F_1$ motor, we generated a difference density map between the cryo-EM density and our built model. Remarkably, clear peaks were seen at the nucleotide binding pockets, indicating that the non-catalytic binding sites in subunit α contained nucleotide, as well as the 'closed' αβ heterodimer (*Figure 5*). These were modeled as ATP and ADP respectively based on known orientations of these nucleotides from high-resolution crystal structures. Interestingly, this did not correspond to the nucleotide binding states of either the *E. coli* (*Cingolani and Duncan, 2011*) or the PS3 crystal structures (*Shirakihara et al., 2015*).

Single particle analysis revealed highly uniform homogeneity of the protein preparation, where 100% of $F_1F_o$ molecules observed demonstrated the extended conformation of εCTD. Our structural data were supported by an enzymatic assay (*Figure 3—figure supplement 2*), where the extent of ATP hydrolysis inhibition of the protein by subunit ε was tested with N,N-dimethyldodecylamine N-oxide (LDAO), a well-known activator of the subunit ε inhibited protein. LDAO used at 0.4% (weight/volume) concentration has been shown to stimulate ATP hydrolysis by the *E coli* protein 3–4

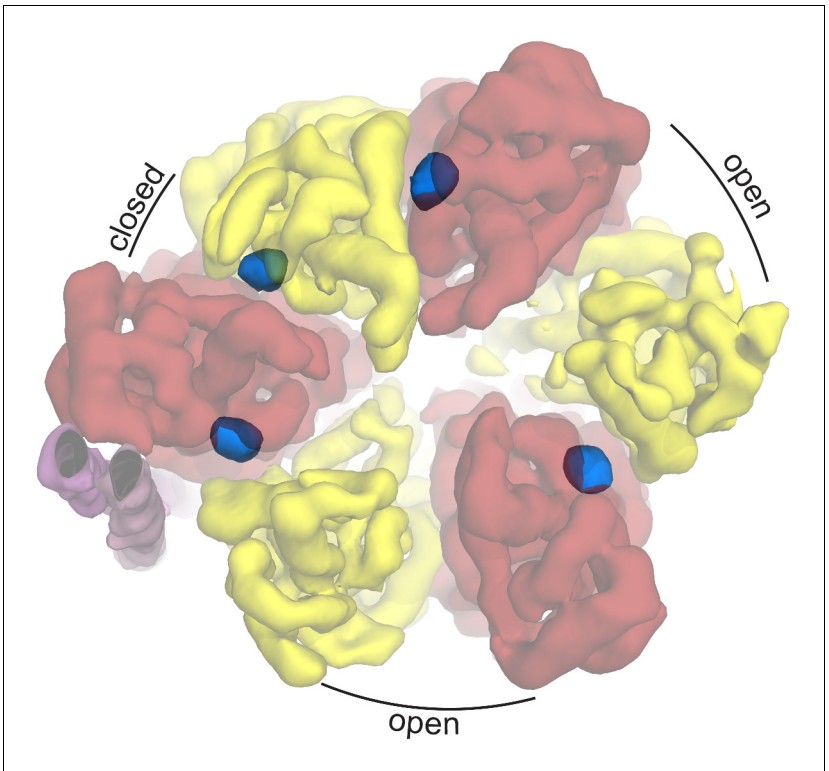

**Figure 5.** Autoinhibted *E. coli* F$_1$-ATPase conformation. The $\alpha\beta$ hetrodimers of state 1 as viewed from the membrane with the peripheral stalk to the left of the figure. The 'open, closed, open' conformation of the F$_1$ motor is labeled and the positions of nucleotides are shown as blue surfaces.

times (*Dunn et al., 1990*; *Peskova and Nakamoto, 2000*), including earlier studies (*Ishmukhametov et al., 2008*, *2016*) on the same cysteine-free F$_1$F$_o$ construct using the same batch of LDAO. Prior to addition of LDAO, the hydrolysis rate was 0.75 µmol ATP/min/mg protein but surprisingly in presence of 0.4% LDAO it was stimulated ~13 times. To our best knowledge, such a high LDAO stimulation of ATP hydrolysis by *E. coli* F$_1$F$_o$ was not described in the literature before and is consistent with our single particle data.

## The *E. coli* F$_o$ motor – architecture of a proton channel

Density in F$_o$ defined the overall architecture of the membrane-embedded motor together with two invaginations of the detergent micelle that have previously been proposed to facilitate proton translocation (*Allegretti et al., 2015*; *Kühlbrandt and Davies, 2016*) (*Figure 6—figure supplement 1*). While the overall density of the *c*-ring was relatively weak, 10 peaks of density were clearly present when viewed from above (*Figure 6—figure supplement 2*), confirming the stoichiometry of the *c*-ring to be decameric in *E. coli* F-ATPase (*Jiang et al., 2001*; *Ballhausen et al., 2009*; *Düser et al., 2009*; *Ishmukhametov et al., 2010*). Furthermore, density inside the *c*-ring corroborates data suggesting it to be filled with phospholipids (*Oberfeld et al., 2006*).

By combining the helical density from the cryo-EM maps (*Figure 6A*), with models previously suggested for the related bovine subunit, together with crosslinking data and transmembrane topography prediction for the *E. coli* F-ATPase (*Jiang and Fillingame, 1998*; *Valiyaveetil and Fillingame, 1998*; *Moore and Fillingame, 2008*; *Wada et al., 1999*), it was possible to build a molecular model of the *a* subunit (*Figure 6B*). The crosslinks mapped to two clusters (*Figure 6—figure supplement 3*), allowing a likely sequence register for the model to be proposed. This was consistent with the two half channel hypothesis, placing Arg210 of subunit *a* adjacent to Asp61 of the *c*-ring (*Figure 6B*). Interestingly, density for the *c* subunit is clearest adjacent to Arg210 of subunit *a* suggesting this area to be well ordered (*Figure 6—figure supplement 4*).

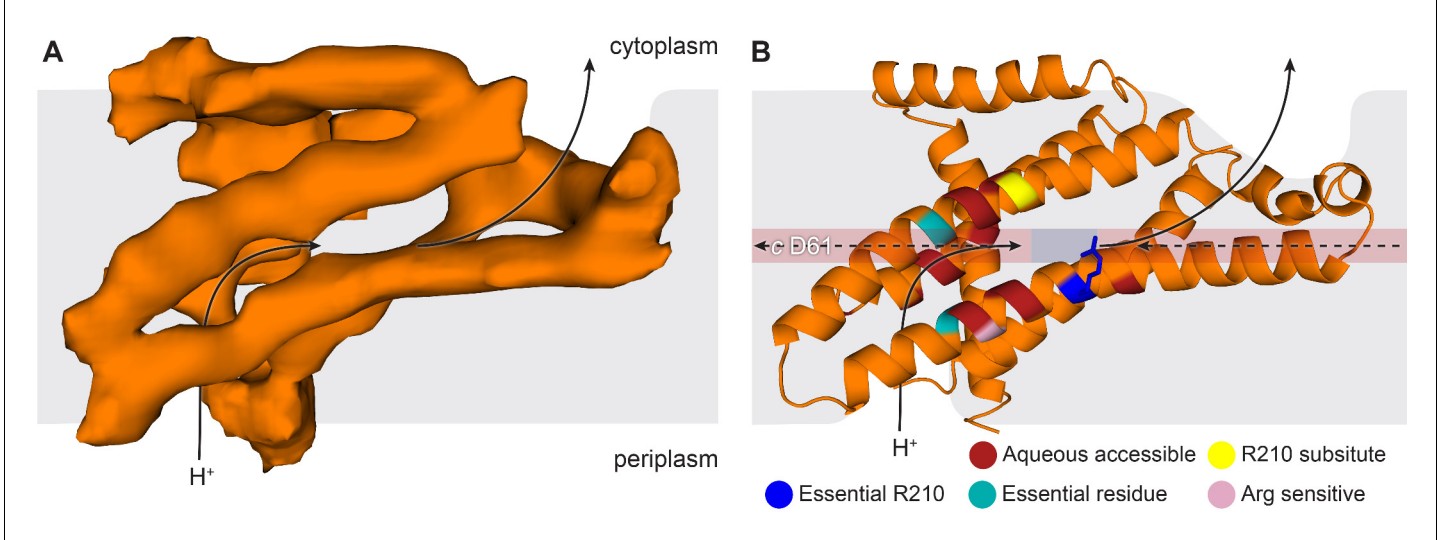

**Figure 6.** The *E. coli* F-ATPase subunit *a* and the suggested path of proton translocation. (**A**) Density map of subunit *a*, shown as orange surface viewed from the *c*-ring. Grey outline depicts invaginations of the detergent micelle, with arrows showing possible proton path. (**B**) Cartoon representation of subunit *a* with a horizontal stripe to depict the position of Asp61 on the *c*-ring (red where Asp61 would be bound to a proton and blue when bound to Arg210). Functional mutants labeled as follows; essential arginine in blue, substitution with Arg210 resulting in functional complex in yellow, mutation to arginine resulting in a dysfunctional complex in teal and residues that are aqueous accessible in red. Solid arrows show a possible proton path via two 'half' channels and dashed arrows show the path when bound to Asp61 of the *c*-ring and rotating.

The following figure supplements are available for figure 6:

**Figure supplement 1.** Aqueous cavities of the *E.coli* F$_O$ motor.

**Figure supplement 2.** View of the State two map from F$_1$ to show *c*-ring stoichiometry (numbered).

**Figure supplement 3.** Crosslinks of the *E.coli* F$_O$ motor.

**Figure supplement 4.** Strong density near Arg210.

**Figure supplement 5.** Functional mutants of *E.coli* F-ATPase subunit *a*.

## Discussion

We have generated cryo-EM maps of a bacterial F-ATPase providing new insights into this rotary ATPase subtype. These maps enabled the generation of a molecular model that presents a framework onto which the vast array of information available on the widely studied *E. coli* enzyme can be mapped, including the attachment of the peripheral stalk to the F$_1$ and F$_o$ motors, the inhibition mediated by the ε subunit, and the stoichiometry of the *c*-ring. In addition, the model confirmed the presence of key features such as the near 'horizontal' helices angled at 20–30° relative to the membrane, indicating that this feature is conserved and is a signature of F, V and A-type ATPases.

Our reconstructions extended previous work (*Wilkens and Capaldi, 1998b*) by showing the structure of the complete peripheral stalk and how it is attached to both the F$_1$ and F$_o$ components. The peripheral stalk functions to counteract rotation of the F$_1$ stator relative to the F$_o$ stator as the central stalk rotates, but must also accommodate conformational changes in the F$_1$ motor during catalysis. The peripheral stalk is based on a long right-handed coiled-coil dimer, that is the hallmark of all rotary ATPase peripheral stalks (*Lee et al., 2010*), showed near parallel α-helices, based on an 11-residue hendecad sequence repeat, spanning the space between the F$_o$ and F$_1$ motors, that changed into a 15-residue quindecad sequence repeat along the F$_1$ motor enabling it to accommodate conformational changes (*Stewart and Stock, 2012*; *Stewart et al., 2012*). Although sequence identity is low (22%), the overall fold of the soluble portion of the peripheral stalk was strikingly

similar to that of *Thermus thermophilus* A-ATPase (*Lee et al., 2010*), illustrating a strong evolutionary pressure for this fold and its function to prevent rotation between the $\alpha\beta$ heterodimers and the *a* subunit while accommodating wobbling of the $F_1$ motor. The bifurcation of the peripheral stalk coiled-coil into two separate helices in the membrane was unexpected, but this arrangement enabled the peripheral stalk to bind to the *a* subunit in two regions. This in turn increased the distance about the fulcrum of interaction, which may help clamp the *a* subunit to the *c*-ring and counteract rotation and pivoting relative to the rotor. The cryo-EM maps also indicated that the peripheral stalk is able to flex about two hinges adjacent to the $F_1$ and $F_o$ motors, enabling it to accommodate conformational changes in the catalytic head.

In addition to its fundamental importance in cell metabolism, the regulation of ATP synthase is also an attractive antibiotic discovery target for pathogenic bacteria closely related to *E. coli* (*Ahmad et al., 2013*). Bacterial F-ATPases employ a unique method of regulation whereby the enzyme can be autoinhibited with the integral subunit $\varepsilon$. In all three rotational states of the *E. coli* F-ATPase, the $\varepsilon$CTD had an extended conformation, albeit with a different proportion of particles observed at each state (46%, 30% and 24%) (*Figure 2—figure supplement 3*), suggesting State one to be the lowest energy. No reconstruction at any stage of data processing contained density corresponding to a closed/down conformation of the $\varepsilon$CTD, and this along with the strong stimulation of ATPase activity by LDAO, suggests that the majority of the protein to be in an autoinhibited form. Although the position of $\varepsilon$CTD relative to the $\alpha\beta$ subunits was similar to that of the mitochondrial inhibitor protein (IF1), the observation that it bound to all three states was different to that seen for IF1 that is bound to a single rotational $F_1$ state ($\alpha/\beta_{DP}$ site proximal to the peripheral stalk) in the $F_1F_o$ ATP synthase dimer structure (*Hahn et al., 2016*). The cryo-EM maps resembled the 'open, closed, open' conformation as seen in the *Bacillus* PS3 $F_1$ crystal structure, despite different nucleotide binding positions. However, the major contacts formed by the $F_1$ motor with the $\varepsilon$CTD in our maps were similar to that of the *E. coli* crystal structure, except that one $\beta$ subunit changed conformation substantially (*Figure 3—figure supplement 1*). Because our cryo-EM study was performed in the absence of externally added nucleotide, it is likely that the structures correspond to the autoinhibited conformations in solution, and the crystal structure of the isolated *E. coli* $F_1$ could instead represent a partially bound state, especially since the crystals were soaked in 1 mM AMPPNP prior to freezing (*Cingolani and Duncan, 2011*).

The *c*-ring is responsible for the rotation of the complex and contains the conserved carboxylate that binds the proton. Different species have varying numbers of subunits in their ring, believed to 'gear' the motor tailoring them to their environment and ranging from 8 to 15 subunits (*Stock et al., 1999*; *Pogoryelov et al., 2007*; *Watt et al., 2010*; *Stewart et al., 2013*). Although the density corresponding to the *c*-ring was quite weak, 10 peaks can be discriminated in the density at either end of the ring (*Figure 6—figure supplement 2*). This confirmed the *c*-ring stoichiometry of *E. coli* F-ATPase that had previously been suggested to be decameric by crosslinking (*Ballhausen et al., 2009*), fusion (*Jiang et al., 2001*) and single molecule analysis (*Düser et al., 2009*; *Ishmukhametov et al., 2010*).

The model of subunit *a* generated from our cryo-EM maps confirmed that the $F_o$ motor likely operates using two half channels separated by a conserved arginine that directs its rotation (*Vik and Antonio, 1994*; *Junge et al., 1997*). Importantly, in this context, our model placed Arg210, which is believed to mediate the rotation of the *c*-ring, adjacent to the conserved carboxylate residue, Asp61, of the *c* subunit that has been shown to bind protons (*Vik and*

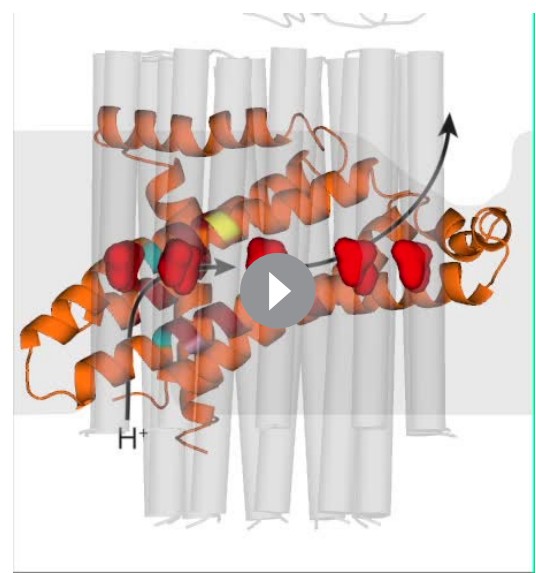

**Video 2.** View of $F_o$ motor during ATP synthesis. Same as main text *Figure 6b*, but with rotating *c*-ring in the foreground.

*Antonio, 1994*; *Pogoryelov et al., 2010*) (*Figure 6b* and *Video 2*). In addition Gln252, which can be substituted with Arg210 and retain ATP synthase function (*Ishmukhametov et al., 2008*; *Hatch et al., 1995*), is positioned on a proximal helix with a similar distance to Asp61 of subunit *c* (*Figure 6B* and *Figure 6—figure supplement 5*). Moreover, mainly charged residues, that have been shown to be aqueous accessible (*Angevine and Fillingame, 2003*; *Angevine et al., 2003*), map to two channel-like areas exposed to solvent by the invagination of the detergent micelle (*Figure 6B* and *Figure 6—figure supplement 5*). Furthermore, residue A217, which has been shown to be sensitive to arginine mutation, and therefore been suggested to be near or part of the aqueous pocket (*Cain and Simoni, 1989*), is positioned next to the periplasmic half channel (*Figure 6B* and *Figure 6—figure supplement 5*). Additionally. residues E219 and H245, which when substituted for one another result in a functional enzyme, are proximal to one another in our model (*Cain and Simoni, 1988*) (*Figure 6—figure supplement 5*). Further analysis of inter subunit crosslinking fits our model well (*Figure 6—figure supplement 3*), with the distances between the c-ring and a subunit being minimal.

In summary, our models show a new level of detail for the bacterial F-ATPase, providing a template for further experiments as well as to guide future antibiotic discovery in related pathogenic bacteria.

## Accession codes

The three models and maps were deposited in the pdb and emDB with codes 5T4O, EMD-8357 (State 1), 5 T4P, EMD-8358 (State 2), 5T4Q and EMD-8359 (State 3).

# Materials and methods

## Protein purification

A cysteine-free version of *E. coli* F-ATPase cloned in plasmid pFV2 and expressed in *E. coli* DK8 strain was used (*Ishmukhametov et al., 2005*). Cells were grown at 37°C in LB medium supplemented with 100 µg/ml ampicillin, for 4–5 hr. The harvested cells were resuspended in lysis buffer containing 50 mM Tris/Cl pH 8.0, 100 mM NaCl, 5 mM $MgCl_2$, 0.1 mM EDTA, 2.5% glycerol and 1 µg/ml DNase I and processed by one pass in French press at 20 kPSI. Cellular debris was removed by centrifuging at 7700 $\times$ g for 15 min, and the membranes were collected by ultracentrifugation at 100,000 $\times$ g for 1 hr. The ATP synthase complex was extracted from membranes at 4°C for 1 hr by resuspending the pellet in extraction buffer consisting of 20 mM Tris/Cl, pH 8.0, 300 mM NaCl, 2 mM $MgCl_2$, 100 mM sucrose, 20 mM imidazole, 10% glycerol, 4 mM digitonin and EDTA-free protease inhibitor tablets (Roche). The complex was then purified by binding on Talon resin (Clontech) and eluted in 150 mM imidazole. The protein was further purified and sugars removed by size exclusion chromatography on a 16/60 Superose six column equilibrated in a buffer containing 20 mM Tris/Cl pH 8.0, 100 mM NaCl, 4 mM digitonin and 2 mM $MgCl_2$. The purified protein was then concentrated to 2 mg/ml for cryo-EM.

## Protein reconstitution into proteoliposomes

Seventy microgram of $F_1F_o$ was reconstituted into extrusion-preformed 100 nm soybean phosphatidylcholine liposomes exactly as descried (*Ishmukhametov et al., 2016*).

## Functional assays

Proton pumping by proteoliposomes was studied using quenching of a pH sensitive fluorescent probe 9-Amino-6-Chloro-2-Methoxyacridine (ACMA) exactly as described (*Ishmukhametov et al., 2016*). The assay was performed with 100 µl of proteoliposomes in 2-ml cuvettes. The reaction was started with 0.25 mM ATP and stopped by 2 µM of the uncoupler FCCP.

ATP hydrolase activity and its stimulation by LDAO was measured with ATP regenerating system using 5 µg of the protein with 1 mM ATP exactly as described (*Ishmukhametov et al., 2016*).

DCCD inhibition of proteoliposomes was done as described (*Ishmukhametov et al., 2016*). DCCD inhibition of pure $F_1F_o$ was done as described (*Ishmukhametov et al., 2005*), with the following modification. Ten microgram of the protein was incubated in 1 ml buffer A (50 mM MES, pH 6.4, 100 mM KCl, 1 mM $MgCl_2$) with 50 µM DCCD for 30 min at room temperature. Control sample

contained 1% ethanol instead of DCCD. Reaction was started by mixing the inhibited protein with 1 ml of buffer A containing all the components of ATP regenerating system.

All the functional experiments presented here were repeated two to three times using the protein isolated by MS and shipped to RI at liquid $N_2$ temperature. Results of typical experiments are shown.

## Cryo-EM grid preparation and data collection

Aliquots of 4 µl of purified *E. coli* F-ATPase at a concentration of 3.58 µM were placed on glow-discharged holey carbon grids (Quantifoils Copper R2/2, 200 Mesh). Grids were blotted for 2 s and flash-frozen in liquid ethane using an FEI Vitrobot Mark IV. Grids were transferred to an FEI Titan Krios transmission electron microscope that was operating at 300 kV. Images were recorded automatically using the FEI EPU software, yielding a pixel size of 1.4 Å. A dose rate of 29 electrons (spread over 20 frames) per Å$^2$ per second, and an exposure time of 2 s were used on the Falcon-II detector. 8640 movies were collected.

## Data processing

MotionCorr (*Li et al., 2013*) was used to correct local beam-induced motion and to align resulting frames. Defocus and astigmatism values were estimated using CTFFIND4 (*Rohou and Grigorieff, 2015*), and 252 micrographs were excluded due to drift or excessive ice contamination. 1208 particles were manually picked and subjected to 2D classification to generate templates for autopicking in RELION (*Scheres, 2012*). The automatically picked micrographs were manually inspected to remove false positives, finally yielding 395,140 particles. These particles then underwent two rounds of 2D classification to generate 22 classes with 311,887 particles. The final particles were classified into four 3D classes using a previously generated model from a low-resolution data set of the same sample (unpublished), low-pass filtered to 60 Å. The resolution was estimated using Fourier Shell Correlation (FSC = 0.143, gold-standard). Three of the four classes containing 104,510 (State 1), 67,829 (State 2) and 53,587 (State 3) particles were movie-refined and post-processed in RELION producing maps at 7.4, 7.8, 8.5 Å, respectively (*Figure 2—figure supplement 10*). State 1 was further processed using masked classification (*Bai et al., 2015*) with residual signal subtraction with a mask created by removing parts of the detergent micelle. Three out of the four classes from this classification containing 95,345 particles were combined and refined to generate the final 6.9 Å map. *Figure 2—figure supplement 3* is a summary of these methods, shown as a flowchart. Local resolution of different parts of the complex was estimated using RELION and ResMap (*Kucukelbir et al., 2014*).

## Model building

Crystal and NMR structures of subunits from *E. coli* ($\alpha\beta\gamma\epsilon$ - 3oaa [*Cingolani and Duncan, 2011*], δ - 2a7u [*Wilkens et al., 2005*], *b* - 1b9u [*Dmitriev et al., 1999*], 1l2p [*Del Rizzo et al., 2002*] and 2khk [*Priya et al., 2009*]) and related organisms (*c* - 3u2f [*Symersky et al., 2012*] and *a* - 5fik [*Zhou et al., 2015*]) were rigid body docked into the highest resolution cryo-EM map and the side chains 'pruned' to Cα. The sequence was mapped to subunit *a* using crosslinks as restraints. Subsequent manual model building and refinement was performed with Coot (*Emsley et al., 2010*), Phenix (*Adams et al., 2010*) and Refmac (*Murshudov et al., 2011*) (excluding subunit *c* due to weak density), with crosslinks again used as external restraints (a summary of the types of models used to build the initial model can be found in *Figure 2—figure supplement 6*). Nucleotide occupancy was determined by first building the model without any nucleotide present, and then segmenting the map and selecting any density with 15% overlap with atoms and deleting this density. Nucleotide was subsequently docked into this difference density using the known positions from previous structures. Once a complete model was built of the highest resolution map, this was docked and refined to the other two maps to create three models. The three models and maps were deposited in the pdb and emDataBank with codes 5T4O, EMD-8357 (State 1), 5 T4P, EMD-8358 (State 2), 5T4Q and EMD-8359 (State 3). Data statistics shown in *Figure 2—source data 1*.

## Acknowledgements

AGS was supported by a National Health and Medical Research Council Fellowship 1090408. DS was supported by National Health and Medical Research Council Fellowships APP1004620 and APP1109961. This work was funded by the National Health and Medical Research Council project grants APP1022143 and APP1047004. ASWW and SS are supported by the Singapore Ministry of Education Academic Research Fund Tier 3 (MOE2012-T3-1-001). R I was supported by BBSRC grant BB/L01985X/1. We thank and acknowledge Daniela Rhodes, the NTU Institute of Structural Biology, the Netherlands Centre for Electron Nanoscopy (NeCEN) at Leiden University, FEI, Sarah Neumann and Max Maletta for help with single particle cryo-EM data collection. NeCEN is supported by the Netherlands Organization for Scientific Research (NWO) and the European Regional Development Fund of the European Commission. The Monash Centre for Electron Microscopy is acknowledged for initial screening of samples.

## Additional information

### Funding

| Funder | Grant reference number | Author |
|---|---|---|
| Biotechnology and Biological Sciences Research Council | BB/L01985X/1 | Robert Ishmukhametov |
| National Health and Medical Research Council | 1004620 | Daniela Stock |
| National Health and Medical Research Council | 1109961 | Daniela Stock |
| National Health and Medical Research Council | 1022143 | Daniela Stock |
| National Health and Medical Research Council | 1047004 | Daniela Stock |
| Ministry of Education - Singapore | MOE2012-T3-1-001 | Sara Sandin |
| National Health and Medical Research Council | 1090408 | Alastair G Stewart |

The funders had no role in study design, data collection and interpretation, or the decision to submit the work for publication.

### Author ORCIDs

Alastair G Stewart, http://orcid.org/0000-0002-2070-6030

## Additional files

### Major datasets

The following datasets were generated:

| Author(s) | Year | Dataset title | Dataset URL | Database, license, and accessibility information |
|---|---|---|---|---|
| Sobti M, Smits C, Wong ASW, Ishmukhametov R, Stock D, Sandin S, Stewart AG | 2016 | Autoinhibited E. coli ATP synthase state 1 | http://www.rcsb.org/pdb/explore/explore.do?structureId=5T4O | Publicly available at the RCSB Protein Data Bank (accession no. 5T4O) |
| Sobti M, Smits C, Wong ASW, Ishmukhametov R, Stock D, Sandin S, Stewart AG | 2016 | Autoinhibited E. coli ATP synthase state 1 | https://www.ebi.ac.uk/pdbe/entry/emdb/EMD-8357 | Publicly available at the Protein Data Bank in Europe (accession no. EMD-8357) |

| Sobti M, Smits C, Wong ASW, Ishmukhametov R, Stock D, Sandin S, Stewart AG | 2016 | Autoinhibited E. coli ATP synthase state 2 | http://www.rcsb.org/pdb/explore/explore.do?structureId=5T4P | Publicly available at the RCSB Protein Data Bank (accession no. 5T4P) |
|---|---|---|---|---|
| Sobti M, Smits C, Wong ASW, Ishmukhametov R, Stock D, Sandin S, Stewart AG | 2016 | Autoinhibited E. coli ATP synthase state 2 | https://www.ebi.ac.uk/pdbe/entry/emdb/EMD-8358 | Publicly available at the Protein Data Bank in Europe (accession no. EMD-8358) |
| Sobti M, Smits C, Wong ASW, Ishmukhametov R, Stock D, Sandin S, Stewart AG | 2016 | Autoinhibited E. coli ATP synthase state 3 | http://www.rcsb.org/pdb/explore/explore.do?structureId=5T4Q | Publicly available at the RCSB Protein Data Bank (accession no. 5T4Q) |
| Sobti M, Smits C, Wong ASW, Ishmukhametov R, Stock D, Sandin S, Stewart AG | 2016 | Autoinhibited E. coli ATP synthase state 3 | https://www.ebi.ac.uk/pdbe/entry/emdb/EMD-8359 | Publicly available at the Protein Data Bank in Europe (accession no. EMD-8359) |

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
