## [Decision Letter]

Thank you for submitting your article "Cryo-EM structures of *E. coli* ATP synthase" for consideration by *eLife*. Your article has been favorably evaluated by Richard Aldrich as the Senior Editor and three reviewers, one of whom, Werner Kühlbrandt (Reviewer #1), is a member of our Board of Reviewing Editors. The following individuals involved in review of your submission have agreed to reveal their identity: John L Rubinstein (Reviewer #2); Thomas Meier (Reviewer #3).

The reviewers have discussed the reviews with one another and the Reviewing Editor has drafted this decision to help you prepare a revised submission.

Summary:

The *E. coli* ATP synthase is an important membrane protein complex that has been extensively characterised by biochemistry and biophysics experiments for decades, yet there has been no structure of this particular ATP synthase to date. All three reviewers agree that this is beautiful work and that the structures have been carefully determined by cryo-EM. In particular, two reviewers comment on the remarkable bifurcation of the *b* subunits within the membrane. Nevertheless, all three reviewers raise essentially the same substantive concern that has to be addressed before the manuscript can be accepted for publication in *eLife*.

Essential revisions:

1) The structure is repeatedly described as an atomic model, which indeed it is. However, anyone can build an atomic model of any protein into a map of any resolution, so the term is meaningless. It is also potentially misleading, because an uninitiated reader might get the impression that this is an atomic resolution structure, which it most definitely is not. It is a map of moderate (~7 Å), resolution. Others have published maps of other ATP synthases at significantly higher resolution, yet have refrained from calling their structures "atomic". The inflationary use of the term "atomic" does not do the cryoEM field or your own research any good.

2) Figure 1, and especially Figure 3 and Figure 4 convey the impression that the map resolution is very much higher than it actually is. The same goes for Figure 2, Figure 5, Figure 2—figure supplement 3 and Video 1. Instead of the experimental map, the authors show spherical atomic surface representations that look like a map at sidechain resolution, or higher. This is not acceptable. What the map really looks like is shown in Figure 1 and Figure 6, which give a true and realistic impression of its quality, which is excellent for 7 Å. Please replace the smooth knobbly atomic model in all figures by the actual map, with a fitted stick model within a transparent volume if you like.

3) Different parts of the mosaic model built into the EM map have very different qualities: some parts are docked crystal structures from *E. coli*, some parts are homology models. Other parts, like the b dimer and the C-terminal domain of δ, are built into the low resolution map directly, while the subunit is based on the model from EM and evolutionary covariance from the bovine enzyme. These latter parts (b, C-terminal domain of δ, a) will be less accurate than the crystal structures or homology models. A better, self-contained, discussion of the variable quality of the different parts of the map, related to how they were derived, is needed.

4) Do the three ATPase models to be deposited in the PDB contain side chains? If so, they should be removed, so as not to mislead future users who will take them at face value for modelling, molecular dynamics, drug binding simulation etc.

5) Subsection “Protein purification”: It is surprising to learn at this late stage, i.e. in the Methods, that the ATP synthase used in this study was in fact a cysteine-free mutant. This important point should be mentioned explicitly in the first part of the Results. How many cysteines were replaced in total? Their positions should be shown in a supplementary figure.

6) How active was the sample used for cryo-EM and was the ATPase activity still coupled? Does the sample activity correspond to the statement (Introduction, last paragraph) that this is an autoinhibited state of the enzyme (so no activity?). The *E. coli* ATP synthase is known to be active only for a short time (hours) in detergent solution, so some biochemical information at the beginning of the Results section would be helpful. Add another supplementary figure to show this if appropriate.

7) Introduction: "Neutralizing this carboxylate enables the c-ring to rotate within the hydrophobic membrane and to access a second aqueous half channel that opens to the cytoplasm into which the protons are released". This describes a key feature of every F_o_ motor. Earlier work (including work by the reviewers) has provided much of the structural and biochemical evidence that supports this statement, and this work should be properly cited – also in the Discussion.

8) Introduction, last sentence: Without further explanation readers will not understand why *E. coli* ATPase can be an antimicrobial target, as *E. coli* is generally known as harmless commensal bacterium.

---

## [Author Response]

*Essential revisions:*

*1) The structure is repeatedly described as an atomic model, which indeed it is. However, anyone can build an atomic model of any protein into a map of any resolution, so the term is meaningless. It is also potentially misleading, because an uninitiated reader might get the impression that this is an atomic resolution structure, which it most definitely is not. It is a map of moderate (~7 Å), resolution. Others have published maps of other ATP synthases at significantly higher resolution, yet have refrained from calling their structures "atomic". The inflationary use of the term "atomic" does not do the cryoEM field or your own research any good.*

We quite accept the comments about the use of “atomic” and have removed this term from the text. The term has either been removed or replaced with “molecular model” when the word “model” seemed ambiguous.

*2) Figure 1, and especially Figure 3 and Figure 4 convey the impression that the map resolution is very much higher than it actually is. The same goes for Figure 2, Figure 5, Figure 2—figure supplement 3 and Video 1. Instead of the experimental map, the authors show spherical atomic surface representations that look like a map at sidechain resolution, or higher. This is not acceptable. What the map really looks like is shown in Figure 1 and Figure 6, which give a true and realistic impression of its quality, which is excellent for 7 Å. Please replace the smooth knobbly atomic model in all figures by the actual map, with a fitted stick model within a transparent volume if you like.*

We have modified the figures and video along the lines suggested. It was not our intention to mislead anyone with the use of a surface representation for the majority of the figures within the manuscript. We had experimented with a range of visual representations for the data, and felt that surface representation gave a very nice figure for illustrative purposes and we had not appreciated how these figures could be misinterpreted. In the light of the reviewers’ comments we have changed all these figures to show either a transparent volume with a cartoon representation fitted (Figure 7) or a segmented map (Figure 7). We tried using the “fitted stick model within a transparent volume” as suggested (Figure 7), but as can be seen, this gave a very confusing illustration (we think the lines add too much detail). Supplements have also been changed and uploaded as part of this resubmission

Author response image 1.**DOI:**
http://dx.doi.org/10.7554/eLife.21598.029

*3) Different parts of the mosaic model built into the EM map have very different qualities: some parts are docked crystal structures from E. coli, some parts are homology models. Other parts, like the b dimer and the C-terminal domain of δ, are built into the low resolution map directly, while the subunit is based on the model from EM and evolutionary covariance from the bovine enzyme. These latter parts (b, C-terminal domain of δ, a) will be less accurate than the crystal structures or homology models. A better, self-contained, discussion of the variable quality of the different parts of the map, related to how they were derived, is needed.*

We are grateful for this suggestion and we have added a section in the Discussion along with a figure supplement that describes the varying quality of the mosaic model.

*4) Do the three ATPase models to be deposited in the PDB contain side chains? If so, they should be removed, so as not to mislead future users who will take them at face value for modelling, molecular dynamics, drug binding simulation etc.*

The three ATPase models deposited in the PDB do not contain side chains, and were truncated to the Cα at the docking/modeling stage. This has also been clarified in the text.

*5) Subsection “Protein purification”: It is surprising to learn at this late stage, i.e. in the Methods, that the ATP synthase used in this study was in fact a cysteine-free mutant. This important point should be mentioned explicitly in the first part of the Results. How many cysteines were replaced in total? Their positions should be shown in a supplementary figure.*

This is a helpful suggestion and we now mention this explicitly at the start of the Results, and once the concept of docking structures into the map is raised, a supplementary figure depicting the positions of the mutated cysteines is cited.

*6) How active was the sample used for cryo-EM and was the ATPase activity still coupled? Does the sample activity correspond to the statement (Introduction, last paragraph) that this is an autoinhibited state of the enzyme (so no activity?). The E. coli ATP synthase is known to be active only for a short time (hours) in detergent solution, so some biochemical information at the beginning of the Results section would be helpful. Add another supplementary figure to show this if appropriate.*

The *E. coli* ATP synthase sample used in this study was purified in a manner previously described and studied (Ishmukhametov et al., 2010) with a different detergent and an additional size exclusion step. No nucleotide was added during purification and therefore the sample was presumed to be active and autoinhibited. However, as the reviewers requested biochemical data, the protein was shipped overseas in a similar manner as for the EM data collection and ATP hydrolysis and proton pumping assays were performed with subsequent addition of DCCD or LDAO. This tested both coupling and the autoinhibition of the complex. This data has been added to the inhibition section of the Results and two additional figure supplements have been added.

We thank the reviewers for this suggestion, which has made this manuscript a far more robust piece of work.

*7) Introduction: "Neutralizing this carboxylate enables the c-ring to rotate within the hydrophobic membrane and to access a second aqueous half channel that opens to the cytoplasm into which the protons are released". This describes a key feature of every F_o_ motor. Earlier work (including work by the reviewers) has provided much of the structural and biochemical evidence that supports this statement, and this work should be properly cited – also in the Discussion.*

Our apologies for not citing this work correctly and have added the following citation on the neutralizing of the carboxylate in the F_o_ complex:

Denys Pogoryelov, Alexander Krah, Julian D Langer, Özkan Yildiz, José D Faraldo-Gómez & Thomas Meier. Microscopic rotary mechanism of ion translocation in the Fo complex of ATP synthases

Nature Chemical Biology 6, 891–899 (2010)

*8) Introduction, last sentence: Without further explanation readers will not understand why E. coli ATPase can be an antimicrobial target, as E. coli is generally known as harmless commensal bacterium.*

While most *E. coli* strains are harmless, some can cause severe foodborne disease (WHO). However, we agree that it’s not at the top of the list when it comes to multidrug resistant pathogenic bacterial. What we implied by this statement was more that understanding the *E. coli* enzyme would broaden our knowledge of the bacterial system, which may be transferable to other bacteria that pose a more substantial risk. The text has been updated to reflect this.